# Baseline Circulating miR-125b Levels Predict a High FIB-4 Index Score in Chronic Hepatitis B Patients after Nucleos(t)ide Analog Treatment

**DOI:** 10.3390/biomedicines10112824

**Published:** 2022-11-05

**Authors:** Jyun-Yi Wu, Yi-Shan Tsai, Chia-Chen Li, Ming-Lun Yeh, Ching-I Huang, Chung-Feng Huang, Jia-Ning Hsu, Meng-Hsuan Hsieh, Yo-Chia Chen, Ta-Wei Liu, Yi-Hung Lin, Po-Cheng Liang, Zu-Yau Lin, Wan-Long Chuang, Ming-Lung Yu, Chia-Yen Dai

**Affiliations:** 1Hepatobiliary Division, Department of Internal Medicine and Hepatitis Center, Kaohsiung Medical University Hospital, Kaohsiung Medical University, Kaohsiung 807, Taiwan; 2Faculty of Internal Medicine and Graduate Institute of Clinical Medicine, College of Medicine, Kaohsiung Medical University, Kaohsiung 807, Taiwan; 3Department of Occupational Medicine, Kaohsiung Medical University Hospital, Kaohsiung Medical University, Kaohsiung 807, Taiwan; 4Health Management Center, Kaohsiung Medical University Hospital, Kaohsiung Medical University, Kaohsiung 807, Taiwan; 5Department of Biological Science and Technology, National Pingtung University of Science and Technology, Pingtung 912, Taiwan; 6School of Medicine, College of Medicine and Center of Excellence for Metabolic Associated Fatty Liver Disease, National Sun Yat-Sen University, Kaohsiung 804, Taiwan; 7College of Professional Studies, National Pingtung University of Science and Technology, Pingtung 912, Taiwan; 8Drug Development and Value Creation Research Center, Kaohsiung Medical University, Kaohsiung 807, Taiwan

**Keywords:** chronic hepatitis B, nucleos(t)ide analogs, miR-125b, hepatitis B virus, biomarker, fibrosis score

## Abstract

The regulatory role of microRNAs (miRNAs) in HBV-associated HCC pathogenesis has been reported previously. This study aimed to investigate the association between serum miR-125b and liver fibrosis progression in chronic hepatitis B (CHB) patients after nucleos(t)ide analog (NA) treatment. Baseline serum miR-125b levels and other relevant laboratory data were measured for 124 patients who underwent 12-month NA therapy. Post-12-month NA therapy, serum miR-125, platelet, AST, and ALT levels were measured again for post-treatment FIB-4 index calculation. Univariate and multivariate logistic regression analyses were performed to identify independent risk factors for a higher post-treatment FIB-4 index. Results showed that baseline miR-125b levels were inversely correlated with the post-treatment FIB-4 index (ρ = −0.2130, *p* = 0.0082). In logistic regression analyses, age (OR = 1.17, *p* < 0.0001), baseline platelet level (OR = 0.98, *p* = 0.0032), and ALT level (OR = 1.00, *p* = 0.0241) were independent predictors of FIB-index > 2.9 post-12-month treatment. The baseline miR-125b level was not significantly associated with a higher post-treatment FIB-4 index (*p* = 0.8992). In 59 patients receiving entecavir (ETV) monotherapy, the alternation of serum miR-125b in 12 months and age were substantially associated with a higher post-treatment FIB-4 index (>2.9), suggesting that miR-125b is a reliable biomarker for detecting early liver fibrosis under specific anti-HBV NA treatments (e.g., ETV).

## 1. Introduction

The prevalence of chronic hepatitis B infection is 4.1%, corresponding to 316 million infected people [1]. It remains an incurable viral disease that can only be controlled by certain medications prescribed in different combinations [2]. There is a significant correlation between hepatitis B viral infection, liver cirrhosis, and hepatocellular carcinoma (HCC). Moreover, hepatitis B virus (HBV) infection and liver fibrosis predispose patients to develop hepatocellular carcinoma (HCC) [3,4]. HCC accounts for 85–90% of all primary liver cancers and is the third most common cause of cancer-related deaths; the five-year survival rate for HCC is 6.9%. The high mortality and poor survival associated with HCC have prompted research on the progression of HBV infection and liver cirrhosis. Liver cirrhosis is characterized by liver tissue fibrosis, which results in the formation of abnormal nodules [5]. As a common consequence of chronic hepatitis B (CHB), it promotes the development of HCC through several possible mechanisms. Molecular mechanisms have been discussed, including identifying HCC driver genes, the p53-RB pathway, and the WNT pathway [6]. Moreover, the long period of hepatic inflammation, as one of the immune responses during chronic HBV infection, increases the hepatocyte turnover rate and gene mutation, which facilitates progression to liver fibrosis, liver cirrhosis, and HCC [3]. Therefore, controlling HBV infection and the early prediction of subsequent possible liver fibrosis are crucial strategies to prevent high-mortality HCC.

Circulating miR-125b levels can inhibit HBV expression in vitro [7] and have long been investigated as an innovative treatment and noninvasive biomarker for CHB, liver fibrosis, and HCC [8,9]. Several studies have illustrated a correlation between miR-125b expression and CHB and its role as a tumor suppressor in several cancers [10,11,12]. Few studies have focused on the role of miR-125b in hepatic inflammation [13]. Zhou et al. demonstrated its ability as a reliable biomarker to predict the virologic response to nucleos(t)ide analog (NA) treatment, which has become the standard of care for patients with CHB [14]. For HCC, studies have described the regulatory role of miRNAs in HBV-associated HCC pathogenesis [8,15]. The ability of miR-125b to act as a novel biomarker for HBV-positive HCC has been demonstrated [16]. Nonetheless, relatively fewer studies have focused on the relationship between miR-125b and liver fibrosis, especially after anti-HBV nucleos(t)ide analog (NA) treatment.

Previous studies have shown that miR-125b can promote hepatic stellate cell activation and liver fibrosis by activating RhoA signaling, and antagonizing miR-125b can significantly alleviate liver fibrosis in CCl4-treated mice [17]. However, there is no corresponding clinical research on the correlation between circulating miR-125b levels and the development of liver fibrosis, especially with anti-HBV NA therapy. The only study that discussed the miR-125b level post-NA therapy mainly focused on HBV/HCV-coinfected patients, rather than HBV infection alone, and there was no evaluation of the correlation between the post-treatment miR-125b level and liver fibrosis [18]. Hence, to further clarify the clinical correlation between circulating miR-125b and the formation of liver fibrosis in post-NA therapy CHB patients, we designed this study to investigate whether baseline and post-treatment serum miR-125b levels can be a reliable predictor of new-onset liver fibrosis after 12-month CHB NA treatment.

## 2. Material and Methods

### 2.1. Study Design

We conducted a retrospective cohort study based on circulating miR125-b levels and serum markers among patients diagnosed with liver fibrosis from stage F0 to F4 after a 12-month NA treatment and patients without a new diagnosis of liver fibrosis after a 12-month NA treatment. The diagnosis and staging of liver fibrosis were made based on the METAVIR score [19], which was applied based on the liver biopsy results, FIB-4 index calculation [20,21], and a FibroScan^®^ (Echosens, Paris, France). We also found an association between serum miR-125b levels and the clinical characteristics of CHB patients, identifying its predictive ability for liver fibrogenesis after NA treatment.

### 2.2. Patients

Patients were recruited from 2004 to 2020 from the medical center of Kaohsiung Medical University Hospital. A total of 127 HBeAg-negative patients underwent NA therapy for HBV infection, according to the Asian-Pacific Association for the Study of the Liver recommendations [22]. The exclusion criteria included patients with other hepatobiliary diseases (e.g., hepatitis C virus (HCV) infection), autoimmune hepatitis, other etiologies of cirrhosis (e.g., primary biliary cirrhosis), primary sclerosing cholangitis, Wilson disease, and α1-antitrypsin deficiency. Three patients were lost to follow-up. Ultimately, 124 patients were included in the analysis. Among the 124 patients, 59 of them received entecavir (ETV) monotherapy, 53 patients received lamivudine (LAM) therapy, and the remaining 12 patients received adefovir, telbivudine (LDT), or tenofovir (TDF) monotherapy or combined therapy. Signed informed consent was obtained from these patients for all interviews, anthropomorphic measurements, blood sampling, and medical record review. This study was approved by the ethics committee of Kaohsiung Medical University Hospital (KMUHIRB-E(II)-20190405).

### 2.3. Laboratory Data

Serum markers, such as aspartate aminotransferase (AST) (Roche Cobas GOT/AST IFCC, Via Casa Sicignano, Sant’Antonio Abate, Italy), alanine transaminase (ALT) (Roche Cobas GPT/ALT IFCC, Via Casa Sicignano, Sant’Antonio Abate, Italy), international normalized ratio (INR), blood urea nitrogen, creatinine, bilirubin, platelets, and albumin, were measured by standard biochemistry tests. HBsAg, HBeAg, and anti-HBe were examined using an enzyme immunoassay (EIA; Abbott Laboratories, North Chicago, IL, USA), and a quantitative HBV DNA analysis was performed using the Roche Cobas Apliprep/Cobas Taqman HBV Test (Roche Molecular System, Roche, Branchburg, NJ, USA).

### 2.4. Extraction of MicroRNAs

The miRNAs were extracted from 200 µL of serum using TRIzol LS (Thermo Scientific, Waltham, MA, USA). The detection of miRNAs was performed by RT-qPCR using TaqMan^®^ MicroRNA assays and measured using a 7900^®^ Sequence Detection System (Applied Biosystems, Lincoln Centre Drive, Foster City, CA, USA). The expression levels of miRNAs in each sample were normalized to the corresponding spike-in cel-39 level [23].

### 2.5. Questionnaire Interview for Patient Profiles

All participants were personally interviewed using structured questionnaires to collect information on demographic characteristics, including alcohol consumption and personal histories of diseases such as previous diagnoses of diabetes, hypertension, heart disease, liver disease, and cardiovascular disease.

### 2.6. Statistical Analysis

We analyzed basic demographic data, including case number, sex, age, body mass index, and the presence of hypertension, diabetes, and metabolic syndrome. Student’s *t*-test, the Mann–Whitney U test, chi-squared test, and Fisher’s exact test were used for statistical tests. All tests were two-sided, and a probability value (*p* value) < 0.05 was considered statistically significant. A 95% confidence interval (CI) for these covariates was calculated. Linear regression analysis was adopted to assess the correlation between variables, as determined by Spearman’s rank correlation coefficient (ρ). Multivariate logistic regression analysis was used to evaluate the odds ratio (OR).

## 3. Results

### 3.1. Characteristics of Patients and Changes in the Parameters

Table 1 summarizes the characteristics of the entire cohort, as well as of the two subgroups divided by the baseline FIB-4 index of 2.9, which has been reported to be the cut-off for advanced fibrosis in patients with CHB [24]. Compared with patients who had a FIB-4 index less than 2.9, those who had a FIB-4 index more than 2.9 were older (53.72 ± 9.76 vs. 43.59 ± 11.26 years, *p* < 0.0001), more likely to have hypertension (28.6% vs. 16.2%, *p* = 0.0226), had higher AST levels (377.9 ± 405.6 vs. 111.7 ± 137.9 U/I, *p* < 0.0001), higher ALT levels (509.1 ± 541.8 vs. 215.2 ± 272.5 U/I, *p* < 0.0001), a lower WBC count (4961.3 ± 1552.9 vs. 6072.5 ± 1753.2 × 10^3^/mm^3^, *p* = 0.0004), a lower platelet count (125.7 ± 41.2 vs. 204.9 ± 70.3 × 10^3^/mm^3^, *p* < 0.0001), a higher bilirubin level (2.69 ± 3.37 vs. 1.37 ± 1.48 mg/dL, *p* = 0.0045), and a lower albumin level (3.72 ± 0.54 vs. 4.26 ± 0.36 gm/dL, *p* < 0.0001). After a follow-up period of 12 months after initial treatment, the alterations in the FIB-4 index and miRNA125b levels are shown in Figure 1. Compared to the baseline FIB-4 index, the FIB-4 index attained after the 12-month treatment demonstrated a significant reduction (*p* < 0.0001). Such a significant difference was not observed regarding mi125b between the baseline and post-treatment levels. The baseline miRNA-125b level and post-treatment FIB-4 indices were significantly and inversely correlated (ρ = −0.2130, *p* = 0.0082) (Figure 2A). However, the alteration in miRNA-125b level during the 12 months was not correlated with the post-treatment FIB-4 index (ρ = −0.1041, *p* = 0.2719) (Figure 2B). Between subgroups divided by a post-treatment FIB-4 index of 2.9, neither baseline miRNA-125b (*p* = 0.2322) nor alteration of miRNA-125b (*p* = 0.2498) showed significant differences (Figure 3).

### 3.2. Baseline and Clinical Parameters Associated with FIB-4 Index after NA Treatment

Table 2 illustrates the results of the univariate logistic regression FIB-4 index analysis. Neither baseline miRNA-125b nor alteration of miRNA-125b during the 12-month treatment period was associated with the risk of the post-treatment FIB-4 index being higher than 2.9. With regard to initial laboratory findings, a higher WBC count (*p* = 0.0156), lower platelet count (< 0.0001), and higher ALT level (*p* = 0.0496) were significantly associated with the risk of a FIB-4 index > 2.9 after 12-month anti-HBV medication treatment. Older age (*p* < 0.0001) was also associated with an increased risk of a FIB-4 index of more than 2.9. Such a significant association was not found for the other variables. Multivariate logistic regression analyses showed that older age (*p* < 0.0001), lower platelet count (*p* = 0.0032), and higher ALT level (*p* = 0.0241), but not lower WBC count, remained significant predictors of a FIB-4 index > 2.9.

### 3.3. miR-125b Predicts Liver Fibrosis Stratified by ETV and LAM Response

Of the 124 analyzed patients, 59 were treated with entecavir and 53 were treated with lamivudine. Table 3 and Table 4 depict the univariate and multivariate logistic regression models in these subgroups. In 59 patients treated with entecavir for 12 months, older age (*p* = 0.0078) and alteration of miRNA-125b levels during treatment (*p* = 0.0157) were substantially associated with a FIB-4 index of more than 2.9 in both univariate and multivariate analyses. A lower baseline platelet level (*p* = 0.0109) was also a risk factor for a high FIB-4 index in the univariate analysis. The baseline miRNA-125b levels were not associated with a high FIB-4 index. In 53 patients treated with lamivudine, older age and lower baseline platelet levels were independently associated with a higher FIB-4 index (≥2.9) in both the univariate and multivariate analyses. Neither baseline miRNA-125b nor alteration of miRNA-125b during treatment was associated with any degree of a high post-treatment FIB-4 index in univariate analyses.

## 4. Discussion

The micro-RNA cluster, miR-125b, is a potential noninvasive biomarker for chronic hepatitis, liver fibrosis, and HCC. In this 1-year retrospective cohort study, we investigated the association between circulating miR-125b levels and liver fibrosis progression in HBV-infected patients treated with oral antiviral medications. We found no significant difference between the miR-125b levels at the baseline and those at the end of the 12th month after initial treatment. Neither the baseline miR-125b level nor miR-125b variation during the 12-month treatment period was associated with a higher FIB-4 index (>2.9). However, baseline miR-125b levels remained significantly and inversely correlated with the FIB-4 index after 12 months of treatment in the overall cohort. The miR-125b variation in 59 patients treated with lamivudine for 12 months was independently associated with a high FIB-4 index (>2.9) risk through univariate and multivariate logistic regression analyses. These real-world findings implicate an association between the alteration of miR-125b levels during certain anti-HBV treatments and possible changes in liver fibrosis.

Different potential noninvasive biomarkers have been investigated for HCC, but they poorly detect early HCC and liver fibrosis. Currently, alpha-fetoprotein is the only reliable marker for diagnosing HCC. Yet, its low specificity, especially in chronic liver disease, limits its application in real-world settings [25]. Given the high correlation between liver cirrhosis and HBV-related HCC, early prediction of liver fibrosis occurrence in HBV patients is an efficient strategy to prevent HBV-related HCC.

Previous studies have shown that the expression levels of miR-125b are serum miRNAs that represent potential biomarkers for hepatocellular carcinoma in patients with chronic hepatitis B virus infection [16,26,27,28,29]. In contrast, the association between miR-125b and early fibrosis has not yet been fully investigated. There is also no evidence demonstrating a correlation between direct-acting HBV treatment and alteration of miR-125b. Giray et al. carried out a study to investigate 24 miRNAs as a potential biomarker for the diagnosis of early liver fibrosis. The study was performed in three patient groups: CHB (*n* = 24), HBV-positive cirrhosis (*n* = 22), and HBV-positive HCC (*n* = 20), along with one control group (*n* = 28). In all these three patient groups, the expression level of miR-125b-5p was reported to be significantly upregulated (1.904–2.854-fold changes) compared to the control group by the Mann–Whitney *U* test.

Nevertheless, when all groups were compared with the control group using a one-way ANOVA test, the expression levels of miR-125b only showed an upregulated tendency without a statistically significant difference (*p* = 0.07192) [16]. Intriguingly, in our study, although the FIB-4 index showed a substantial decrease after 12 months of HBV treatment, the alteration in miR-125b was not significant. Between the two subgroups, divided by a post-treatment FIB-4 index of 2.9, neither the baseline miR-125b level nor the alternation showed significant differences. Our findings indicate that NA therapy for CHB might prevent the progression of liver fibrosis by counteracting upregulated miR-125b levels.

In line with our hypothesis, Yeh et al. carried out a prospective single-center study to investigate the outcomes of HBV infection in 79 HBV/HCV-coinfected patients after 1 year of DAA therapy [18]. This study reported reduced serum levels of miR-125b in HBV/hepatitis C virus (HCV)-coinfected patients after direct antiviral agents (DAAs) were administered for HCV infection and a continuous decline in miR-125b levels after stopping DAAs in patients with HBV reactivation, compared with no change in miR-125b levels in patients without HBV reactivation. This partially explains the miR125b alternation in our study as not only related to the progression of liver fibrosis but also to viral replication and involving more complex virus–host interactions, which require further investigation.

Zhou et al. conducted a retrospective cohort study to investigate the predictive value of baseline miR-125b levels for NA therapy in patients with CHB. A total of 66 HBeAg-positive CHB patients who had received LDT-optimized therapy (*n* = 39) or TDF monotherapy (*n* = 27) for 144 weeks were analyzed. Their results indicated that the baseline serum miR-125b level is an independent predictor of a complete response (defined as HBV DNA < 500 IU/mL and HBeAg seroconversion). This finding suggests that baseline miRNA-125b is a satisfying biomarker for HBeAg seroconversion following 144-week NA treatment [14].

In the present study, although baseline miR-125b was not an independent predictor for a FIB-4 index higher than 2.9 after 12 months of NA treatment in the overall HBV-positive cohort, there was a significant inverse correlation between the baseline serum miR-125b and post-treatment FIB-4 index. After we further analyzed 59 patients treated with LAM monotherapy for 12 months, the multivariate logistic regression model confirmed the substantial association between serum miR-125b alteration (OR = 0.22, *p* = 0.0157) and the risk of a higher FIB-4 index. In the same analysis, the baseline miR-125b level showed borderline significance (*p* = 0.0727) in univariate analyses. The major difference between our study and that of Zhou et al. is the duration of NA treatment. With a one-year therapy shorter than the duration of Zhou et al.’s study, the association between serum miR-125b alteration and fibrosis has been elucidated. Further studies are needed to validate the findings concerning long-term NA therapy.

After further interpretation of the comparison between 59 patients receiving ETV and 53 patients receiving LAM, we found that older age was a common independent predictor of a higher post-treatment FIB-4 index in both treatment groups and all cohorts. In all 124 patients, lower baseline platelet levels and higher baseline ALT levels were also significantly associated with the risk of a higher post-treatment FIB-4 index in both univariate and multivariate analyses.

However, the baseline serum ALT level was no longer a significant predictor in either the ETV or LAM group. The baseline platelet level remained an independent risk factor in the LAM group for a higher FIB-4 index after 12 months of treatment. Instead of platelet levels, the alteration of miR-125b during 12-month treatment can substantially and independently predict a higher post-treatment FIB-4 index, as discussed above. These findings suggest that the risk factors for liver fibrosis progression may differ between different NA treatment groups. Circulating miR-125b, a noninvasive and an easily accessible molecule, can be a reliable biomarker for predicting early liver fibrosis in HBV-positive patients, particularly those receiving specific NA treatment, such as ETV. Further large-scale studies with longer follow-up periods are needed to validate this finding.

This study had some limitations. First, it was a retrospective study conducted in a single center, although there has been no previous retrospective or prospective multicenter study with the same study design. Second, we did not generally define liver cirrhosis with a liver biopsy, given the standardized strategies that aim to prevent unnecessary healthcare-associated risks. Instead, we analyzed the FIB-4 index, validated in various patient settings with patient age and laboratory profiles [22,30,31,32,33]. Third, our analysis did not include some liver-disease-relevant data, such as INR, due to incomplete collection. However, data analyzed in a real clinical setting remain crucial in CHB and liver fibrosis.

In conclusion, the baseline serum miR-125b level was significantly and inversely correlated with the post-treatment FIB-4 index. Furthermore, the alteration in miR-125b levels during specific NA therapy, such as ETV, may be associated with the FIB-4 index after 12-month treatment. These findings are fundamental in recognizing miR-125b as a biomarker for the early progression of liver fibrosis during HBV treatment, especially the widely adopted ETV. Additionally, given that the long-term goal of surveillance for liver cirrhosis and HCC is to prevent the occurrence of early liver fibrosis, instead of monitoring miR-125b alone, our findings merit further studies to validate the role of different NA treatments in affecting serum miR-125b levels and their relationship to liver fibrosis.

## Figures and Tables

**Figure 1 biomedicines-10-02824-f001:**
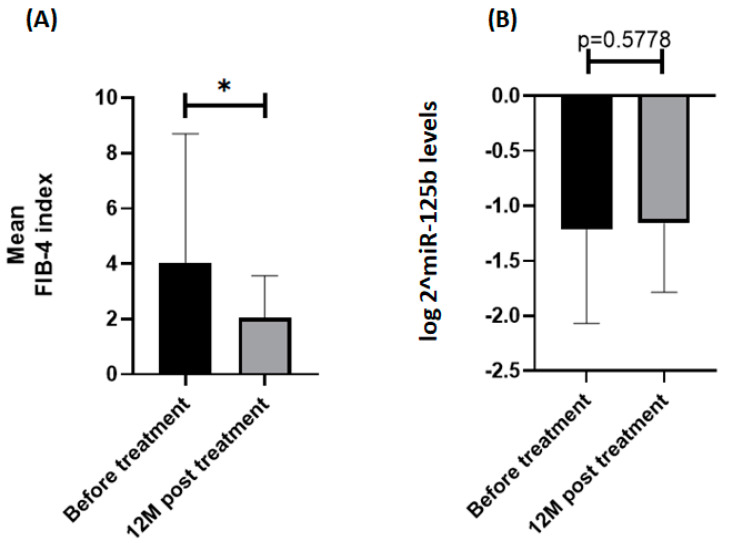
Comparing the differences in the (**A**) mean FIB−4 index and (**B**) log 2^miR−125b levels at the baseline and post−treatment. Error bars show mean ± standard deviation. *p* value from the Mann–Whitney test is shown. * = *p* < 0.05.

**Figure 2 biomedicines-10-02824-f002:**
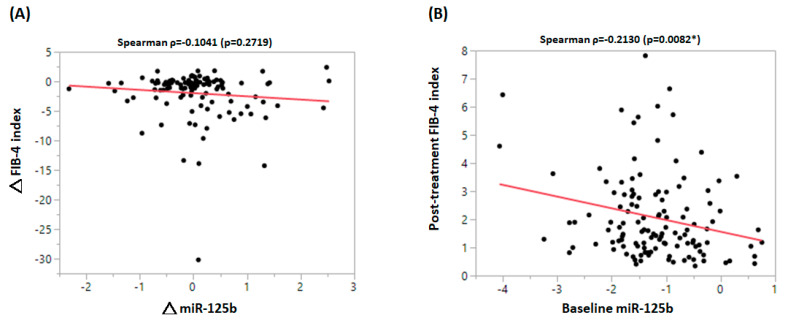
Linear regression analysis was used to assess the correlation between (**A**) the change in miRNA−125b and the change in the FIB−4 index, and (**B**) the baseline miRNA−125b (serum miRNA−125b, i.e., log 2^baseline miR−125b) and post-treatment FIB−4 index. △FIB−4 index, 12-month FIB−4 index; △miRNA−125b, log 2^−delta post−miR−125b^ − log 2^−delta pre−miR−125b^. * = *p* < 0.05.

**Figure 3 biomedicines-10-02824-f003:**
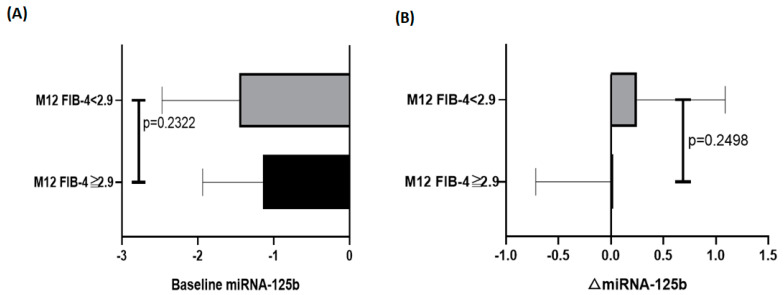
Comparing two subgroups divided by a post-treatment FIB−4 index of 2.9, showing the differences in the (**A**) baseline miRNA−125b (log 2^baseline miR−125b) and (**B**) △miRNA−125b (log 2^−delta post−miR−125b^ − log 2^−delta pre−miR−125b^). Error bars show mean ± standard deviation. *p* value from the Mann–Whitney test is shown.

**Table 1 biomedicines-10-02824-t001:** Patient’s characteristics.

Variable	Overall	FIB-4 Score ≤ 2.9	FIB-4 Score > 2.9 (*n* = 54)	*p* Value
	(*n* = 124)	(*n* = 70)		
Mean age (years)	47.95 (11.75)	43.49 (11.26)	53.72 (9.76)	<0.0001 *
Gender				0.3810
Male	95	52	43	
Female	29	18	11	
BMI (kg/m^2^)	23.99 (3.59)	24.10 (3.37)	23.88 (3.87)	0.7620
BH (cm)	166.59 (8.02)	168.23 (7.60)	164.52 (8.13)	0.0060 *
BW (kg)	66.75 (12.20)	68.25 (11.22)	64.92 (13.18)	0.0746
HBV DNA > 2000 IU/mL (%)	76.61	70.00	85.19	0.0476 *
HBeAg(+)	30.89 (38/123)	31.88 (22/69)	29.63 (17/54)	0.7883
Lab data (mean, SD)				
WBCs (×10^3^/mm^3^)	5585.79 (1751.43)	6072.50 (1753.19)	4961.32 (1552.85)	0.0004 *
Platelet (×10^3^/mm^3^)	170.43 (71.12)	204.93 (70.29)	125.70 (41.18)	<0.0001 *
AST (U/L)	227.64 (314.82)	111.69 (137.87)	377.94 (405.57)	<0.0001 *
ALT (U/L)	343.16 (435.35)	215.17 (272.48)	509.07 (541.77)	0.0001 *
Creatinine (mg/dL)	0.89 (0.48)	0.86 (0.24)	0.92 (0.66)	0.5363
Bilirubin (mg/dL)	2.00 (2.57)	1.37 (1.48)	2.69 (3.37)	0.0045 *
Albumin (gm/dL)	4.02 (0.53)	4.26 (0.36)	3.72 (0.54)	<0.0001 *
HB (g/dL)	13.95 (1.56)	14.20 (1.60)	13.64 (1.46)	0.0512
Log 2^−delta miRNA 125b^	−1.21 (0.85)	−1.20 (0.84)	−1.22 (0.88)	0.8992
Ct values of cel-39 (internal control; mean ± SD)	27.35 (1.65)	27.34 (1.74)	27.36 (1.55)	0.9673
Comorbidities	(*n* = 98)	(*n* = 56)	(*n* = 42)	
Diabetes mellitus (%)	10.20	8.93	11.90	0.6300
Hypertension (%)	17.34	8.93	28.57	0.0110 *
	(*n* = 122)	(*n* = 68)	(*n* = 54)	
Alcohol use (%)	16.40	16.18	16.67	0.9421

BMI, body mass index; BH, body height; BW, body weight; WBC, white blood cell count; PLT, platelet count; AST, aspartate aminotransferase; ALT, alanine aminotransferase; Cr, creatinine; HB, hemoglobin level; delta miRNA 125b represents Ct _miR−125b_-Ct _cel−39_. All values are expressed as the mean (standard deviation (SD)). The *p* value was calculated for the continuous variables using Student’s *t*-test or the Mann–Whitney test, and the χ2 test was used for the categorical variables, with α = 0.05. * = *p* < 0.05.

**Table 2 biomedicines-10-02824-t002:** Predictor of 12M FIB-4 index analysis.

	Univariate Analyses		Multivariate Analyses	
Variable	OR (95%CI)	*p* Value	OR (95%CI)	*p* Value
Age	1.17 (1.09–1.25)	<0.0001 *	1.17 (1.09–1.26)	<0.0001 *
Gender (male/female)	0.89 (0.33–2.37)	0.8188		
HBV DNA > 2000 IU/mL	2.11 (0.67–6.69)	0.2034		
HBeAg(+)	0.41 (0.14–1.17)	0.0963		
Lab data				
WBCs	1.00 (1.00–1.00)	0.0156 *	1.00 (1.00–1.00)	0.1562
Platelet	0.98 (0.97–0.99)	<0.0001 *	0.98 (0.96–0.99)	0.0032 *
AST	1.00 (1.00–1.00)	0.7116		
ALT	1.00 (1.00–1.00)	0.0496 *	1.00 (1.00–1.00)	0.0241 *
Creatinine	0.87 (0.31–2.45)	0.7900		
Bilirubin	1.08 (0.93–1.25)	0.3138		
Albumin (gm/dL)	0.51 (0.23–1.15)	0.1058		
HB (g/dL)	0.77 (0.58–1.01)	0.0601		
Log 2^−delta pre−miR−125b^	0.65 (0.40–1.08)	0.0938		
Δ Log 2^−delta miRNA 125b^	0.65 (0.37–1.13)	0.1270		
Comorbidities				
Diabetes mellitus	1.37 (0.32–5.76)	0.6699		
Hypertension	1.91 (0.62–5.88)	0.2596		
Alcohol use	0.86 (0.80–0.26)	0.8019		

BMI, body mass index; BH, body height; BW, body weight; WBC, white blood cell count; PLT, platelet count; AST, aspartate aminotransferase; ALT, alanine aminotransferase; Cr, creatinine; HB, hemoglobin level; delta miRNA 125b represents Ct _miR−125b_-Ct _cel−39_; Δlog 2^−delta miR−125b^ = Log 2^−delta post−miR−125b^ − Log 2^−delta pre−miR−125b^. * = *p* < 0.05.

**Table 3 biomedicines-10-02824-t003:** Baseline vs. 12M FIB-4 index analysis in ETV (*n* = 59).

	Univariate Analyses		Multivariate Analyses	
Variable	OR (95%CI)	*p* Value	OR (95%CI)	*p* Value
Age	1.14 (1.05–1.25)	0.0033 *	1.17 (1.04–1.32)	0.0078 *
Gender (male/female)	1.27 (0.30–5.42)	0.7442		
HBV DNA > 2000 IU/mL	1.76 (0.33–9.32)	0.5036		
HBeAg(+)	0.19 (0.02–1.65)	0.1332		
Lab data				
WBCs	1.00 (1.00–1.00)	0.2872		
Platelet	0.98 (0.97–0.99)	0.0109 *	0.99 (0.98–1.00)	0.1522
AST	1.00 (1.00–1.00)	0.6767		
ALT	1.00 (1.00–1.00)	0.2354		
Creatinine	0.89 (0.32–2.44)	0.8213		
Bilirubin	1.06 (0.90–0.12)	0.4909		
Albumin (gm/dL)	0.42 (0.13–1.31)	0.1362		
HB (g/dL)	0.66 (0.42–1.03)	0.0718		
Log 2^−delta pre−miR−125b^	0.52 (0.25–1.06)	0.0727		
Δ Log 2^−delta miRNA 125b^	0.34 (0.13–0.88)	0.0268 *	0.22 (0.06–0.75)	0.0157 *
Comorbidities				
Diabetes mellitus	1.38 (0.30–6.36)	0.6756		
Hypertension	2.00 (0.48–8.30)	0.3400		
Alcohol use	1.27 (0.33–4.95)	0.7314		

BMI, body mass index; BH, body height; BW, body weight; WBC, white blood cell count; PLT, platelet count; AST, aspartate aminotransferase; ALT, alanine aminotransferase; Cr, creatinine; HB, hemoglobin level; delta miRNA 125b represents Ct _miR−125b_-Ct _cel−39_; Δ Log 2^−delta post−miR−125b^ − Log 2^−delta pre−miR−125b^. * = *p* < 0.05.

**Table 4 biomedicines-10-02824-t004:** Baseline vs. 12M FIB-4 index analysis in LAM (*n* = 53).

	Univariate Analyses		Multivariate Analyses	
Variable	OR (95%CI)	*p* Value	OR (95%CI)	*p* Value
Age	1.18 (1.06–1.32)	0.0030 *	1.22 (1.06–1.41)	0.0068 *
Gender (male/female)	0.22 (0.04–1.05)	0.0579		
HBV DNA > 2000 IU/mL	1.22 (0.22–6.84)	0.8222		
HBeAg(+)	0.82 (0.17–3.86)	0.8030		
Lab data				
WBCs	1.00 (1.00–1.00)	0.0503		
Platelet	0.97 (0.95–0.99)	0.0094*	0.95 (0.90–1.00)	0.0314 *
AST	1.00 (1.00–1.00)	0.9120		
ALT	1.00 (1.00–1.00)	0.1204		
Creatinine	0.59 (0.03–13.72)	0.7413		
Bilirubin	1.17 (0.82–0.12)	0.3856		
Albumin (gm/dL)	0.48 (0.12–1.93)	0.3026		
HB (g/dL)	0.74 (0.46–1.19)	0.2139		
Log 2^−delta pre-miR-125b^	0.82 (0.34–1.97)	0.6508		
Δ Log 2^−delta miRNA 125b^	1.10 (0.45–2.71)	0.8338		
Comorbidities				
Diabetes mellitus	1.10 (0.30–6.36)	0.9945		
Hypertension	0.90 (0.08–9.97)	0.9316		
Alcohol use	1.50 (0.33–4.95)	0.9980		

BMI, body mass index; BH, body height; BW, body weight; WBC, white blood cell count; PLT, platelet count; AST, aspartate aminotransferase; ALT, alanine aminotransferase; Cr, creatinine; HB, hemoglobin level; delta miRNA 125b represents Ct _miR−125b_-Ct _cel−39_; Δ Log 2^−delta post−miR−125b^ − Log 2^−delta pre−miR−125b^. * = *p* < 0.05.

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
