# Peer review of "Baseline Circulating miR-125b Levels Predict a High FIB-4 Index Score in Chronic Hepatitis B Patients after Nucleos(t)ide Analog Treatment"

_biomedicines, 2022, doi:10.3390/biomedicines10112824_

Round 1

Reviewer 1 Report

A nice study measuring circulating miR-125 levels to predict high Fib-4 index in chronic hepatitis B patients. However, there are some issues with this study that should be addressed.

1. Methodology. RNA was extracted from serum OR plasma. Were both used? Was plasma available for some patients and serum for others? They should be separated in analysis if the latter.

2. Two really crucial study limitations: no availability or bilirubin levels and biopsy at the diagnosis (the gold standard).

3. The novelty of the study. It has been already published that upregulated miR-125b expression is an indicator of chronic hepatitis (independent of viral replication). 

4. Figure titles Fig 1 and Fig 4 requires re-wording.

Author Response

A nice study measuring circulating miR-125 levels to predict high Fib-4 index in chronic hepatitis B patients. However, there are some issues with this study that should be addressed.

  1. Methodology. RNA was extracted from serum OR plasma. Were both used? Was plasma available for some patients and serum for others? They should be separated in analysis if the latter.

We appreciate your comment and the accompanying questions. We extracted all RNA samples from serum. Further, we have deleted “or plasma” from line 126 to avoid any misunderstanding.

  1. Two really crucial study limitations: no availability or bilirubin levels and biopsy at the diagnosis (the gold standard).

We thank you for the comment. Regarding the bilirubin level, we had analyzed the bilirubin levels of patients, and we have added it as one of our variables in the article.

We totally agree with your valuable comment regarding the gold standard role of liver biopsy. However, clinically, biopsy can be scarcely used in routine practice for liver cirrhosis diagnosis. Instead, non-invasive diagnostic score and formula are evolving as safer, cheaper, better tolerated diagnostic modalities for such patients and a better way to track the dynamics of fibrosis progression or regression. Although FIB-4 score is not the best or the gold standard for diagnosis, it has been widely validated and can enhance the generalizability of our study in real practice.

  1. The novelty of the study. It has been already published that upregulated miR-125b expression is an indicator of chronic hepatitis (independent of viral replication). 

We thank you for your crucial comment. We have cited that previous study on the role of miR-125b as a biomarker. However, that study focused mainly on HBV-HCC which means their target population was patients with late stage of liver fibrosis that is irreversible. However, our study explored miR-125b as a biomarker to predict early fibrosis, before the occurrence of Liver cirrhosis or even HCC.

  1. Figure titles Fig 1 and Fig 4 requires re-wording.

We thank you for your comment. Accordingly, we have re-worded the Fig legends appropriately.

Reviewer 2 Report

Wu and colleagues are interested in the relationship between miR-125b levels and FIB-4 index score in chronic hepatitis B patients after specific anti-HBV NAs treatment. They demonstrated that the alternation of serum miR-125b in 12 months is inversely and significantly associated with post-treatment higher FIB-4 index in 59 patients treated with LAM, which could be used as a biomarker in detecting early liver fibrosis. Overall, the study is advantageous for the noninvasive diagnosis of liver fibrosis, despite several non-neglectable limitations as outlined by the authors. I have the following comments for the authors' consideration:

1.     My primary concern is: since the FIB-4 index is calculated according to the formula FIB-4 = Age (years)×AST (U/L)/[PLT(109/L)×ALT1/2 (U/L)], it is reasonable that age, PLT, and ALT are independent predictors of FIB-index > 2.9 after 12-month treatment. What is the important point of this conclusion in this study?

2.     Abstract, last sentence: is there any hint that miR-125b can be a potential therapeutic target against chronic hepatitis B derived from this study? If none, please omit or soften this description.

3.     Materials and Methods 2.5: the structured questionnaires contain demographic characteristics, including education level, occupation, cigarette smoking duration and frequency, alcohol consumption, and personal histories of diseases. However, these data have not been presented in the results section.

4.     Results 3.3: is there any plausible explanation that the alteration of miRNA-125b levels during treatment was substantially associated with a FIB-4 index of more than 2.9 in 59 patients treated with lamivudine; in contrast, the alteration of miRNA-125b during treatment was not associated with any degree of high post-treatment FIB-4 index in 53 patients treated with entecavir. These two results seem conflicting.

5.     Results 3.3: I wonder why were the 12 patients respectively received Adefovir (ADV), Telbivudine (LDT), or Tenofovir (TDF) monotherapy or combine-therapy not analyzed or presented here? Is it due to the minimal number?

6.     Discussion lines 304–305: this conclusion is apparently overstated since it is true only under specific NAs treatment such as LAM.

7. In Figure 1, I suggest adding panels (A) and (B) to be consistent with Figure 4. Furthermore, Figures 2 and 3 could be combined into a single Figure.

8.     In the maintext, there are numerous grammatical errors that should be avoided carefully. Proofreading by an English native speaker is recommended.

Minor points:

1.     Line 28: change “viral” to “virus”

2.     Line 46: “LAM” should be abbreviated after lamivudine in line 43

3.     Line 80: define “NAs”

4.     Line 85: delete “also”

5.     Line 108: …hepatitis C virus…

Author Response

Open Review 2

Wu and colleagues are interested in the relationship between miR-125b levels and FIB-4 index score in chronic hepatitis B patients after specific anti-HBV NAs treatment. They demonstrated that the alternation of serum miR-125b in 12 months is inversely and significantly associated with post-treatment higher FIB-4 index in 59 patients treated with LAM, which could be used as a biomarker in detecting early liver fibrosis. Overall, the study is advantageous for the noninvasive diagnosis of liver fibrosis, despite several non-neglectable limitations as outlined by the authors. I have the following comments for the authors' consideration:

  1. My primary concern is: since the FIB-4 index is calculated according to the formula FIB-4 = Age (years)×AST (U/L)/[PLT(109/L)×ALT1/2(U/L)], it is reasonable that age, PLT, and ALT are independent predictors of FIB-index > 2.9 after 12-month treatment. What is the important point of this conclusion in this study?
    We thank you for this critical question. Age, PLT, AST, and ALT are baseline data; however, the post 12-month treatment age, PLT, AST, ALT  are components of the formula. This study elucidates how baseline age, PLT, AST and ALT level can predict post-treatment 12-month FIB-4 index. Whether baseline age, PLT, AST, ALT can predict post-treatment FIB-4 index or not was questionable before our analysis. However, our analysis depicts that not all of these four baseline components are significantly associated with 12M FIB-4 index.
  2. Abstract, last sentence: is there any hint that miR-125b can be a potential therapeutic target against chronic hepatitis B derived from this study? If none, please omit or soften this description.

We thank you for the comment. Based on your suggestion, we have modified the description.

  1. Materials and Methods 2.5: the structured questionnaires contain demographic characteristics, including education level, occupation, cigarette smoking duration and frequency, alcohol consumption, and personal histories of diseases. However, these data have not been presented in the results section.

We appreciate your point of view. We have removed the items that were not eventually analyzed.

  1. Results 3.3: is there any plausible explanation that the alteration of miRNA-125b levels during treatment was substantially associated with a FIB-4 index of more than 2.9 in 59 patients treated with lamivudine; in contrast, the alteration of miRNA-125b during treatment was not associated with any degree of high post-treatment FIB-4 index in 53 patients treated with entecavir. These two results seem conflicting.

We appreciate your comment.

First of all, we sincerely apologize for the huge error that we made during data analysis. We accidentally reversed the name of ETV and LAM in our statistics software. That is, in the ETV group (n=59) the alteration of miR-125b levels was significantly associated with FIB-4 index value greater than 2.9, and not in the LAM group (n=53).

Regarding the issue, when comparing the ETV and LAM groups, the only significant difference between them was the post-treatment FIB-4 score. ETV: 2.40 (0.20) vs. LAM: 1.74 (0.21), p value=0.0225.

ETV group revealed higher degree of fibrosis after the 12-month NAs therapy. We mentioned in the Discussion section (lines 236-238) that the miR125b alternation in our study was not only related to the progression of liver fibrosis but also to the viral replication and involves more complex virus-host interactions, which require further investigation. Based on the different outcomes seen in the ETV and LAM groups, different correlation between the alteration of miRNA-125b levels during treatment and FIB-4 index greater than 2.9 can be expected, though the detailed reasons need to be further clarified through future studies.

  1. Results 3.3: I wonder why were the 12 patients respectively received Adefovir (ADV), Telbivudine (LDT), or Tenofovir (TDF) monotherapy or combine-therapy not analyzed or presented here? Is it due to the minimal number?

Thank you for this interesting and important question. Yes, ETV and LAM, each have 59 and 53 samples, which are similar. On the other hand, ADV, LDT, and TDF monotherapy or combine-therapy were administered in only a total of 12 patients; the individual number of patients for each therapy was too small to be compared and analyzed.

  1. Discussion lines 304–305: this conclusion is apparently overstated since it is true only under specific NAs treatment such as LAM.

We very appreciate your comment. To be more specific, we have revised “NAs therapy” to ETV therapy.

  1. In Figure 1, I suggest adding panels (A) and (B) to be consistent with Figure 4. Furthermore, Figures 2 and 3 could be combined into a single Figure.

We sincerely thank you for your suggestions. We have made revisions based on your suggestions.

  1. In the main text, there are numerous grammatical errors that should be avoided carefully. Proofreading by an English native speaker is recommended.

We appreciate your comment and thank you for pointing this out. The revised manuscript has been proofread by native English speakers.

Minor points:

  1. Line 28: change “viral” to “virus”
  2. Line 46: “LAM” should be abbreviated after lamivudine in line 43
  3. Line 80: define “NAs”
  4. Line 85: delete “also”
  5. Line 108: …hepatitis C virus…

We sincerely thank you for pointing out these inadvertent errors. We have corrected all of these errors in the revised manuscript.

Round 2

Reviewer 1 Report

I would like to thank The Authors for addressing suggested changes. All appropriate changes were made. Manuscript reads much clearer. Overall quality is significantly improved. 

Reviewer 2 Report

I am satisfied with the modifications made by the authors and have no further critical comments.